# *Mettl5* coordinates protein production and degradation of PERIOD to regulate sleep in *Drosophila*

Xiaoyu Wu[1†], Xingzhuo Yang[1†], Tiantian Fu[1†], Yikang Rong[2], Juan Du[1*]

[1]State Key Laboratory of Agricultural and Forestry Biosecurity, MOA Key Lab of Pest Monitoring and Green Management, Department of Entomology, College of Plant Protection, China Agricultural University, Beijing, China; [2]MOE Key Lab of Rare Pediatric Diseases, Hengyang College of Medicine, University of South China, Hengyang, China

*For correspondence:
dujuan9981@cau.edu.cn

[†]These authors contributed equally to this work

## eLife Assessment

The authors present **useful** findings demonstrating that the RNA modification enzyme Mettl5 regulates sleep in *Drosophila*. Through transcriptome- and proteome-wide analyses, the authors identified downstream targets affected in heterozygous mutants and proposed that Mettl5 regulates the translation and degradation of clock genes to maintain normal sleep function. Through additional analyses, the authors provided **solid** evidence supporting this model.

**Abstract** Sleep plays a critical role in animal physiology, primarily governed by the brain, and its disruption is prevalent in various brain disorders. Mettl5 is associated with intellectual disability (ID), which often includes sleep disturbances. However, the mechanism underlying these sleep disruptions in ID remains poorly understood. In this study, we investigated the sleep phenotypes resulting from *Drosophila Mettl5* mutations. Rescue experiments revealed that *Mettl5* functions predominantly within neurons and glia marked by *Mettl5*-Gal4 to regulate sleep. Previous work established that Mettl5 forms a complex with Trmt112 to influence rRNA methylation. Notably, a mutation in *Trmt112* recapitulated these sleep disturbances, implicating translational regulation by the Mettl5/Trmt112 complex. Subsequent RNA-seq and Ribo-seq analyses of *Mettl5[1bp]* mutants uncovered downstream effects, including altered expression of proteasome components and clock genes. Rescue experiments confirmed that the net increase in PERIOD protein underlies the sleep phenotype. This study illuminates the interplay between ribosome function, clock genes, and the proteasome in sleep regulation, highlighting the integrated roles of protein synthesis and degradation. These findings could potentially provide an example for in vivo study of rRNA methylation function, expand our understanding of protein homeostasis in sleep, and offer insights into the sleep phenotypes associated with ID.

## Introduction

Sleep is essential for animal physiology, and understanding its molecular mechanisms has significant implications for both basic research and clinical applications. Although previous studies have identified key sleep regulators (*Du et al., 2021*), fundamental questions remain regarding the roles of cellular processes such as protein synthesis and degradation in sleep regulation.

Growing evidence suggests a dynamic interplay between sleep regulation and protein homeostasis. Protein synthesis is particularly active during sleep (*Lyons et al., 2023*), while sleep deprivation affects

translational initiation (*Costa et al., 2019*). The proteasome, a critical mediator of ubiquitin-dependent protein degradation, regulates synapse homeostasis by modulating ribosomal components (*Costa et al., 2019*), thereby influencing synaptic function. Intriguingly, proteasome components exhibit oscillating expression patterns at the transcriptional level in human cells (*Desvergne et al., 2016*). A recent study in *Drosophila* showed that mutation in a proteasome component alters sleep patterns (*Fernández-Cruz et al., 2020*). Despite these advances, the precise mechanisms by which protein synthesis and degradation influence sleep, and how these processes interact, remain unresolved.

While several factors influencing proteasome assembly and subunit transcription have been identified (*Kapetanou et al., 2022*), the translational control of proteasome subunits remains poorly understood. Additionally, proteasome inhibition has been shown to alter ribosome function (*Galimberti et al., 2016*; *Costa et al., 2019*; *Palanca et al., 2014*), raising the question of whether the proteasome itself is regulated by the status of the translation machinery. Exploring this relationship could provide key insights into the interplay between proteasome activity and translation.

In *Drosophila*, *Mettl5* exhibits 18S ribosomal RNA $N^6$-methyladenosine (m⁶A) methyltransferase activity (*Leismann et al., 2020*) and interacts with Trmt112 to facilitate this function, a mechanism conserved in humans (*Leismann et al., 2020*; *van Tran et al., 2019*). Although *Mettl5* depletion abolishes m⁶A modification on 18S rRNA without impairing rRNA maturation, it affects fly orientation behavior. In mammalian cells, *Mettl5*-mediated 18S rRNA m⁶A modification regulates stem cell fate determination and neural function (*Wang et al., 2022*). However, it remains unclear whether *Mettl5* modulates global or transcript-specific translation profiles in vivo, particularly in tissues rather than cell lines. Moreover, while *Mettl5* is linked to intellectual disability (ID) with comorbid sleep disturbances, its mechanistic role in these disorders is unknown. Testing whether *Mettl5* loss affects behaviors like sleep in *Drosophila* could clarify this connection.

We discovered that *Mettl5* regulates sleep in *Drosophila*. To investigate the underlying mechanism, we performed RNA-seq and Ribo-seq on *Mettl5* mutants, revealing dysregulation of multiple clock genes and proteasome components. This suggests *Mettl5* coordinates protein production and degradation, which are crucial for protein homeostasis. Follow-up experiments confirmed that the protein level of Period was upregulated in *Mettl5* mutants, contributing to the sleep phenotype. This study highlights that ribosome defects can perturb proteasome function, uncovering a mechanism that couples protein degradation with synthesis. Additionally, by mapping the genome-wide downstream gene profile of an rRNA methylation modifier, this study offers insights into the gene-specific roles of ribosome function.

## Results

### *Mettl5* regulates sleep in *Drosophila*

*Mettl5* contains a predicted $N^6$ adenine-specific nucleic acids methyltransferase domain. In order to study the function of this gene in sleep regulation, we generated CRISPR–Cas9 knockout mutants. We created two alleles. *Mettl5^1bp^*, which deleted 1 bp in the coding region, results in a truncated version (*Figure 1A*) and downregulation of *Mettl5* at the mRNA level (*Figure 1B*). Another allele, *Mettl5^9bp^*, which results in three amino acids deleted version of *Mettl5* (*Figure 1A*), does not cause significant change at the mRNA level (*Figure 1B*). We found that heterozygous *Mettl5^1bp^* mutants exhibited significantly reduced nighttime sleep (*Figure 1C–G*), particularly during early night (ZT12–ZT16; *Figure 1C*). Quantitative analysis revealed increased wakefulness at ZT14 in mutants (*Figure 1H*). 'awake %' was used to indicate the percentage of awake fruit fly population at specific time points (e.g., ZT14). The quantitative nighttime sleep latency measurements indicated a delayed sleep start in mutants (*Figure 1—figure supplement 1G*). In addition, *Mettl5* expression was significantly upregulated during the sleep recovery period after mechanical sleep deprivation (*Figure 1I, J*). *Mettl5* mutant displayed significantly increased sleep rebound in 24 hr after sleep deprivation (*Figure 1K, L*), indicating its effects on sleep homeostasis. Moreover, results of sleep arousal assay at ZT19 indicated that the percentage of aroused flies is significantly more than the control group (*Figure 1M*). These results demonstrate that the truncated *Mettl5* mutation causes sleep deficits, establishing *Mettl5* as a novel sleep regulator in *Drosophila*.

To confirm the specificity of the sleep phenotype in *Mettl5^1bp^*, we performed genetic rescue experiments. Introduction of a single wild-type *Mettl5* copy completely rescued both the reduced sleep

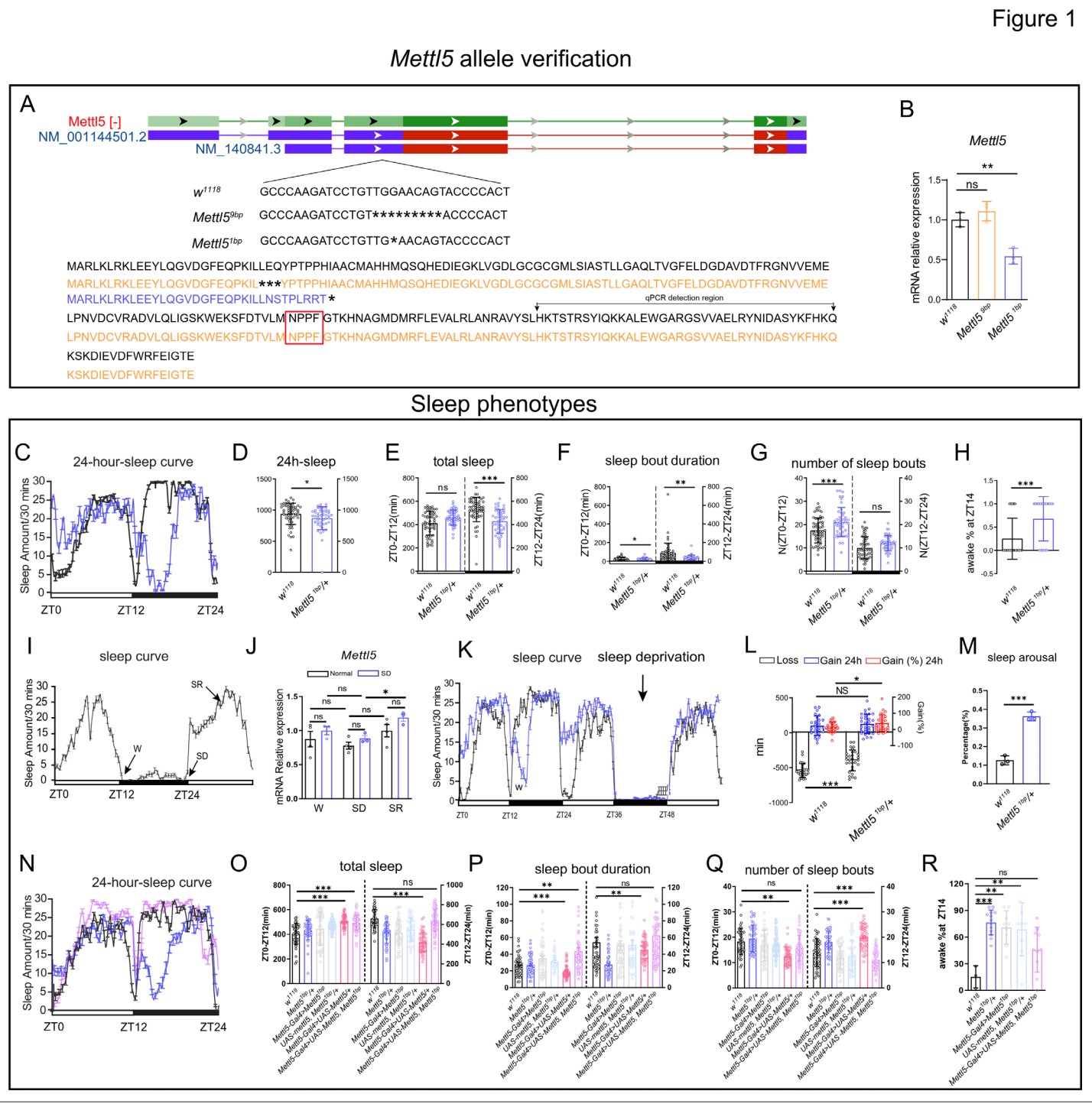

**Figure 1.** *Mettl5* is a regulator of *Drosophila* sleep. (**A**) Diagram illustrating CRISPR–Cas knockout of 1 or 9 bases in the *Mettl5* gene. The corresponding protein sequence is listed with the predicted $N^6$ adenine-specific nucleic acids methyltransferase domain highlighted in the red box. (**B**) Relative expression of *Mettl5* mRNA in homozygous *Mettl5^{1bp}* and *Mettl5^{9bp}* mutant male flies compared to control flies. (**C**) Sleep curve throughout the day for *Mettl5* mutant male flies (blue) and control flies (black). (**D**) Total sleep of *Mettl5* mutant male flies and control flies in 24 hr. (**E**) Total sleep of *Mettl5* mutant male flies and control flies within day and night, respectively. (**F**) Sleep bout duration of *Mettl5* mutant male flies and control flies. (**G**) Number of sleep bouts of *Mettl5* mutant male flies and control flies. (**H**) Percentage of awake for *Mettl5* mutant flies and control flies. (**I**) Sleep curve is tracked throughout the entire day prior to sleep deprivation and during the daytime sleep rebound period. (**J**) *Mettl5* mRNA expression level at different time points. W (wake), SD (sleep deprivation), SR (sleep recovery). (**K**) Sleep curve is tracked throughout the entire day prior to sleep deprivation and during the daytime sleep rebound period in *Mettl5* mutant male flies (blue) and control flies (black). (**L**) Response to sleep deprivation and performance

*Figure 1 continued on next page*

*Figure 1 continued*

measures in *Mettl5* mutants and controls. Black bars represent the amount of sleep lost during the 24-hr sleep deprivation period, blue bars indicate the amount of sleep regained, whereas the red bars indicate the proportion of sleep recovered (right *y*-axis). (**M**) Sleep arousal of *Mettl5[1bp]* male flies and control flies at ZT19. (**N**) Sleep curve throughout the day for the following genotypes: *w[1118]* (black), *Mettl5[1bp]/+* (blue), and *Mettl5-Gal4, UAS-Mettl5, Mettl5[1bp]/+* (pink). (**O**) Total sleep of the indicated genotypes. (**P**) Sleep bout duration of the indicated genotypes. (**Q**) Number of sleep bouts of the indicated genotypes. (**R**) Percentage of awake for the indicated genotypes. For * stands for $p < 0.05$, ** stands for $p < 0.01$, *** stands for $p < 0.001$, ns stands for not significant. For letter-based annotations, groups with no significant differences share the same letter; groups with significant differences are assigned new letters.

The online version of this article includes the following figure supplement(s) for figure 1:

**Figure supplement 1.** *Mettl5* functions in a portion of neurons and glia.

amount and increased wakefulness at ZT14 (*Figure 1N–R*), demonstrating that these phenotypes specifically result from *Mettl5* deficiency.

The observed expression pattern of *Mettl5* further supports its sleep regulatory function. Using *Mettl5*-Gal4 reporter lines, we found expression in both neurons (colocalizing with Elav staining; *Figure 1—figure supplement 1A–C*) and glial cells (colocalizing with REPO staining; *Figure 1—figure supplement 1D–F*). Behavioral analyses revealed additional mutant phenotypes consistent with sleep dysregulation. We tested the climbing ability of *Mettl5* mutation, which showed an increase (*Figure 1—figure supplement 1H*). RNAi knocking down of *Mettl5* showed a consistent phenotype of downregulated sleep amount during the nighttime (*Figure 1—figure supplement 1I–L*).

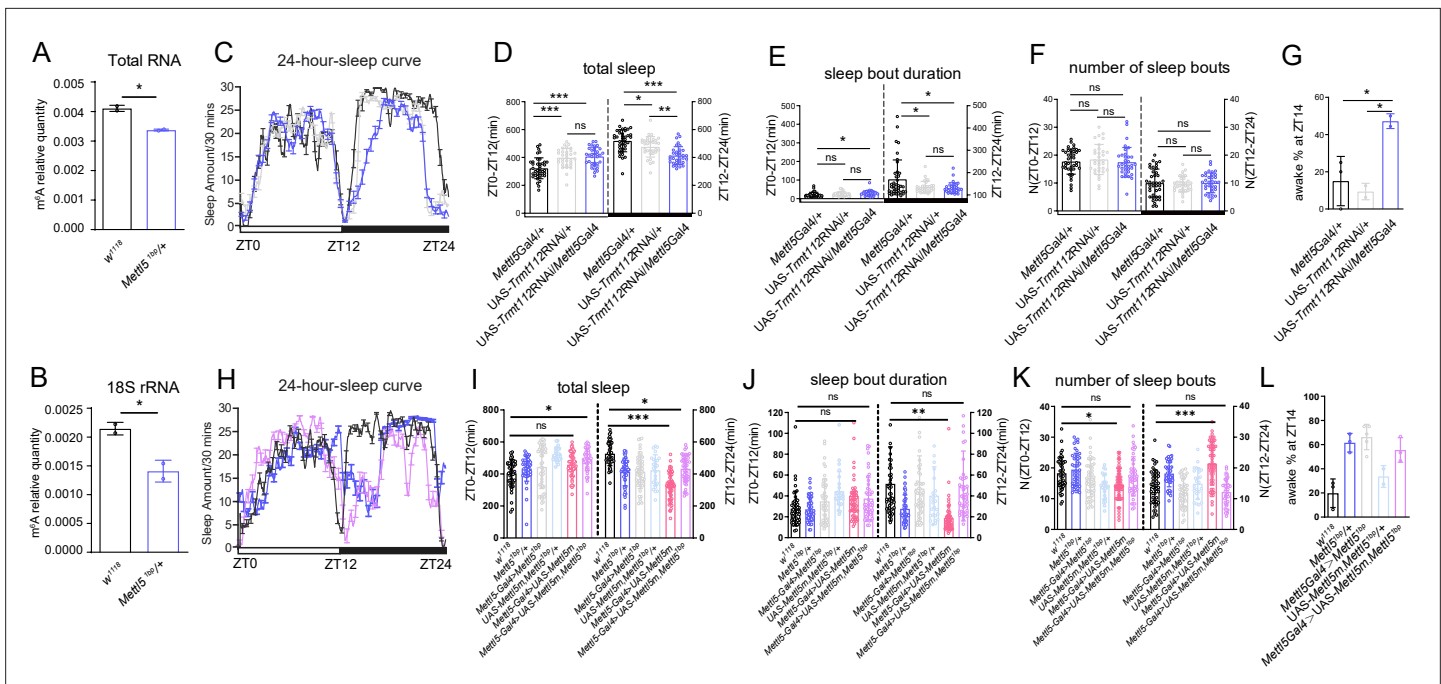

**Figure 2.** *Mettl5* regulation of *Drosophila* sleep was dependent on its methyltransferase activity. (**A**) The m⁶A level in the total RNA of *Mettl5* mutant male flies. (**B**) The m⁶A level in the 18S rRNA of *Mettl5* mutant male flies. (**C**) The sleep curve throughout the day shows the sleep pattern of induced *Trmt112* RNAi male flies and control flies. (**D**) Total sleep of induced *Trmt112* RNAi male flies and control flies. (**E**) Sleep bout duration in induced *Trmt112* RNAi male flies and control flies. (**F**) Number of sleep bouts in induced *Trmt112* RNAi male flies and control flies. (**G**) Percentage of awake in *Trmt112* RNAi and control flies. (**H**) Sleep curve throughout the day for *Mettl5* mutant male flies, induced *Mettl5m* overexpression male flies, and control flies. (**I**) Total sleep of *Mettl5* mutant male flies, induced *Mettl5m* overexpression male flies, and control flies. (**J**) Sleep bout duration in *Mettl5* mutant male flies, induced *Mettl5* overexpression male flies, and control flies. (**K**) Number of sleep bouts in *Mettl5* mutant male flies, induced *Mettl5* overexpression male flies, and control flies. (**L**) Percentage of awake in *Mettl5* mutant male flies, induced *Mettl5* overexpression male flies, and control flies. For * stands for $p < 0.05$, ** stands for $p < 0.01$, *** stands for $p < 0.001$, ns stands for not significant. For letter-based annotations, groups with no significant differences share the same letter. Groups with significant differences are assigned new letters.

## *Mettl5* regulates *Drosophila* sleep through its methyltransferase activity

Previous studies established that *Mettl5* interacts with *Trmt112* to regulate 18S rRNA m⁶A modification (*Leismann et al., 2020*; *van Tran et al., 2019*). Consistent with these findings, our LC–MS/MS analysis revealed significantly reduced m⁶A levels in both total RNA and 18S rRNA from heterozygous *Mettl5^{1bp}* (*Figure 2A, B*). To determine whether Mettl5's sleep regulatory function depends on its methyltransferase activity, we performed two key experiments. First, we examined the phenotype of *Trmt112* knockdown. We found that the *Mettl5*-Gal4 driven *Trmt112* RNAi resulted in a similar phenotype to *Mettl5^{1bp}* (*Figure 2C–G*), suggesting that *Mettl5* regulates sleep through its 18S rRNA m⁶A modification activity. The efficiency of the *Trmt112* RNAi line was tested in a previous study (*López-Varea et al., 2021*). More importantly, we performed a rescue experiment with a mutated form of *Mettl5* that lacks the NPPF amino acids required for its methyltransferase activity (*Figure 2A*; *Iyer et al., 2016*). We observed that this mutant *Mettl5* failed to rescue the sleep phenotype caused by *Mettl5^{1bp}* (*Figure 2H–L*). Together, these results demonstrate that *Mettl5*'s methyltransferase activity is essential for its role in sleep regulation, likely through its function in 18S rRNA m⁶A modification.

## RNA-seq and Ribo-seq revealed the downstream gene profile of *Mettl5*

To better understand the downstream events of *Mettl5*, we performed RNA-seq and Ribo-seq to assess transcriptomic and translational changes in *Mettl5^{1bp}*. Principal Coordinates Analysis of RNA-seq and Ribo-seq datasets revealed clear separation between mutant and control groups (*Figure 3—figure supplement 1A, B, D, E*). High reproducibility was found among biological replicates (*Figure 3—figure supplement 1C, F*), confirming data quality. Transcriptome analysis identified 1053 significantly differentially expressed genes ($|\log_2(\text{fold change})| \geq 1$ and p.adj <0.05), comprising 217 upregulated and 836 downregulated transcripts (*Figure 3A, B*). Parallel ribosome profiling revealed 299 translationally regulated genes ($|\log_2(\text{fold change})| \geq 0.265$ and p.adj <0.05), with 149 upregulated and 150 downregulated targets compared to the controls (*Figure 3C*). Heatmaps of the top 100 differentially expressed genes from both RNA-seq and Ribo-seq analyses were generated according to the rlog transformed values (*Figure 3—figure supplement 2A, B*).

To explore the biological implication of differentially expressed genes at transcriptional and translational levels, we performed Gene Ontology (GO) enrichment analysis. At the transcriptional level (*Figure 3—figure supplement 3A, C*), the most significantly enriched biological processes included cellular response to chemical stimulus, small molecule biosynthetic process, and fatty acid metabolic process (*Figure 3—figure supplement 3A*), with supporting gene networks (*Figure 3—figure supplement 3C*). Translational-level analysis (*Figure 3—figure supplement 3B, D*) revealed prominent enrichment for organic acid and amino acid metabolic processes, lipid catabolism, cellular respiration, and transport mechanisms. Notably, we identified strong associations with circadian regulation, including circadian regulation of gene expression and entrainment of the circadian clock.

The GO results usually contain a long list of enriched terms that have highly redundant information and are difficult to summarize. So, we performed a simplified enrichment analysis that visualizes the summaries of clusters by word cloud for the GO enrichment result at transcriptional and translational levels. For transcriptionally significant DEGs (*Figure 3—figure supplement 4A*), major enriched processes included metabolic and stimulus–response pathways, wing disc and imaginal system development, ion-channel homeostasis and transport, and cytokinesis/cell cycle transitions. Translationally regulated genes additionally showed enrichment for sleep–wake cycles, circadian behavior, and neuronal cell death (*Figure 3—figure supplement 4D*). Cellular component analysis highlighted chromosomal structures, membrane-bound complexes, and vesicles (*Figure 3—figure supplement 4B*), while molecular functions predominantly involved enzymatic activities, ion/protein binding, receptor activities, and transcription processes (*Figure 3—figure supplement 4C*). The strong concordance between RNA-seq and Ribo-seq enrichment profiles (*Figure 3—figure supplement 4E–F*) further validated these findings.

Our gene set enrichment analysis (GSEA) identified pathways enriched in differentially expressed genes at both transcriptional and translational levels. To capture subtle but biologically important changes that might be excluded by stringent statistical thresholds, we analyzed gene sets ranked by $\log_2$ fold change without p-value filtering. The GSEA–GO and GSEA–KEGG results, sorted by normalized enrichment score, revealed several key pathways (*Figure 3—figure supplement 5A–D*).

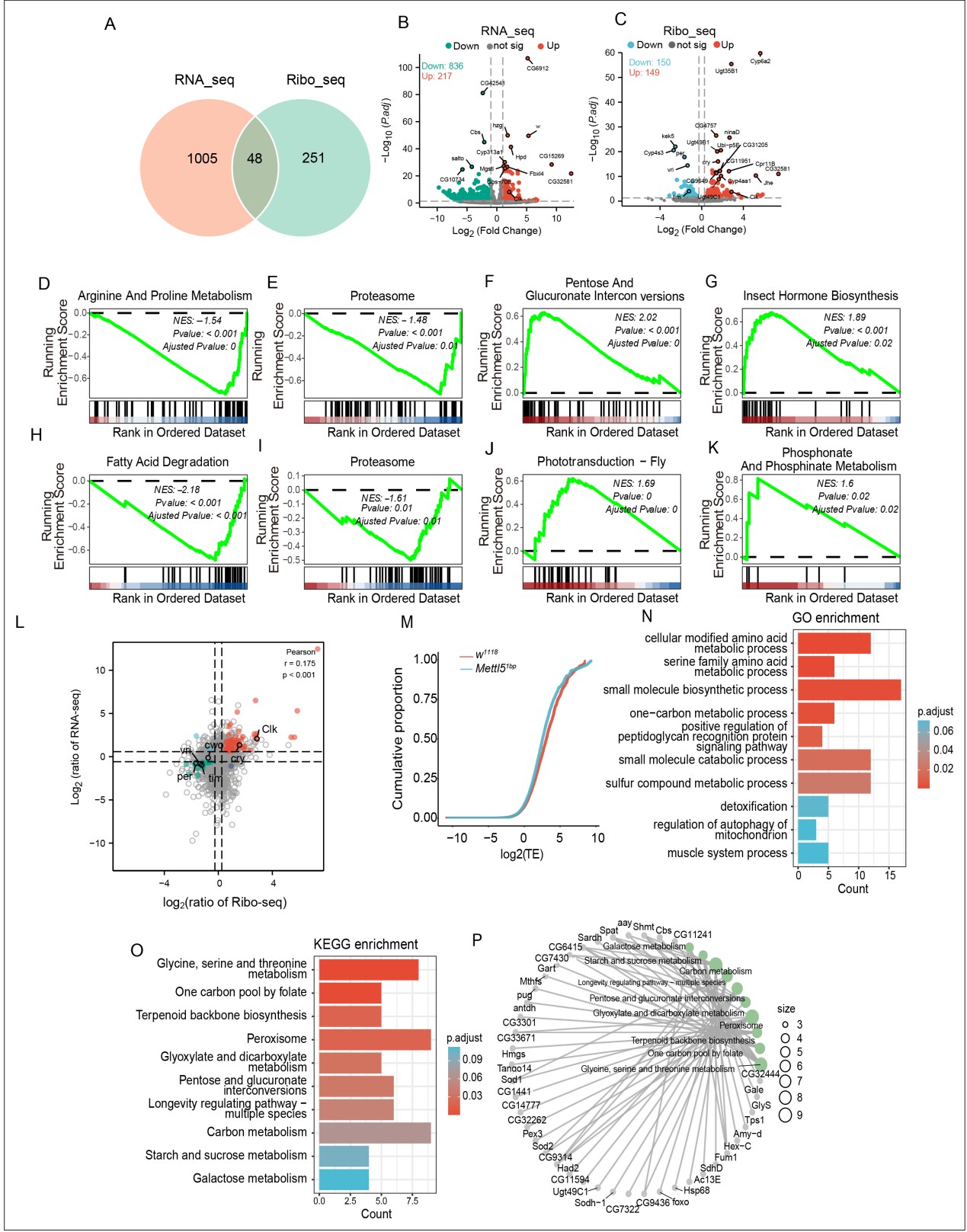

**Figure 3.** RNA-seq and Ribo-seq analysis revealed changes in the gene profile of *Mettl5^1bp*. (**A**) Venn diagram depicting the number of significant differentially expressed genes revealed by RNA-seq and Ribo-seq. (**B**) Volcano plot representing the differentially expressed genes identified by RNA-seq. Genes that met the criteria of |log$_2$(fold change)| ≥1 and p.adjust <0.05 were considered significantly expressed, marked in orange for downregulation and green for upregulation, comparing with the controls. (**C**) Volcano plot representing the differentially expressed genes identified by

*Figure 3 continued on next page*

*Figure 3 continued*

Ribo-seq. Candidates that satisfied the criteria of |log$_2$(fold change)| ≥0.265 and p.adjust <0.05 were regarded as significantly expressed, marked in red for downregulation and blue for upregulation, respectively. Gene set enrichment analysis of differentially expressed genes revealed by RNA-seq (**D–G**) and Ribo-seq (**H–K**). All the plots are generated using the KEGG gene set database. The bar chart at the bottom of each panel shows the distribution of target genes for each pathway according to their rank position. Each vertical line represents a gene. Genes on the left show positive correlation with *Mettl5$^{1bp}$*, while genes on the right show negative correlation with *Mettl5$^{1bp}$*. The green line indicates the enrichment score (ES), and NES stands for normalized enrichment score. (**L**) Distribution of the differentially expressed genes revealed by both RNA-seq and Ribo-seq. (**M**) Cumulative distribution of translation efficiency (TE) frequencies among *w$^{1118}$* and *Mettl5$^{1bp}$*. (**N, O**) Gene Ontology (GO) and KEGG enrichment of significantly changed TE-related genes between *w$^{1118}$* and *Mettl5$^{1bp}$*. The color of the bar indicates the enrichment p.adjust value. (**P**) KEGG network showing the top 10 pathways and associated genes. The size of the dots represents the number of genes in the pathway.

The online version of this article includes the following figure supplement(s) for figure 3:

**Figure supplement 1.** Quality control of RNA-seq and Ribo-seq samples.

**Figure supplement 2.** Heatmap of differentially expressed genes between *Mettl5$^{1bp}$* and *w$^{1118}$*.

**Figure supplement 3.** Enrichment analysis of differentially expressed genes between *Mettl5$^{1bp}$* and *w$^{1118}$*.

**Figure supplement 4.** Simplified Gene Ontology (GO) enrichment analysis for differentially expressed genes in RNA-seq and Ribo-seq.

**Figure supplement 5.** Gene set enrichment analysis (GSEA) for differentially expressed genes found in RNA-seq and Ribo-seq.

**Figure supplement 6.** Gene Ontology (GO) and KEGG enrichment analysis of genes that were significantly changed at both transcriptional and translational levels.

**Figure supplement 7.** Translation efficiency (TE) correlation and P-site offsets.

**Figure supplement 8.** Global scan of ribosome-protected fragments (RPFs) distribution.

**Figure supplement 9.** Ribo-seq has revealed that the *Mettl5$^{1bp}$* leads to changes in global translation features.

Notably, the proteasome pathway showed significant suppression at both transcriptional and translational levels (p.adjust <0.05) (***Figure 3E, I***). Transcriptional-level analysis highlighted disruptions in arginine and proline metabolism and insect hormone biosynthesis (p.adjust <0.05) (***Figure 3D, G***). While, while translational changes predominantly affected fatty acid degradation and phosphonate/phosphinate metabolism (p.adjust <0.05) (***Figure 3H–K***).

To better understand the correlation between transcription and translation levels, we performed correlation analysis of differentially expressed genes based on fold change of DEGs (***Figure 3L***). 977 genes showed opposing trends between transcription and translation, while only 24 genes exhibited concordant changes. We identified 240 translation-specific and 3309 transcription-specific DEGs, with each cluster undergoing separate enrichment analysis (***Figure 3—figure supplement 6A–H***). Notably, core clock genes (*Clk*, *tim*, etc.) displayed significant changes at both levels (***Figure 3L***).

To isolate translational effects, we calculated translation efficiency (TE) differences between *Mettl5$^{1bp}$* and *w$^{1118}$* controls (***Figure 3M***). Among 1,204 genes with significantly altered TE, GO enrichment highlighted amino acid metabolism and small molecule biosynthesis (***Figure 3N***), while KEGG analysis emphasized glycine/serine/threonine metabolism and one-carbon pool by folate pathways (***Figure 3O, P***).

## Ribo-seq revealed that *Mettl5$^{1bp}$* led to changes of some global translation features

Using Ribo-seq, we compared global translation features between the *w$^{1118}$* and *Mettl5$^{1b}$* groups. The TE correlation coefficients among the three biological replicates of each genotype ranged from 0.87 to 0.93, demonstrating high reproducibility within genotypes (***Figure 3—figure supplement 7A–F***). We found that the length of ribosome-protected RNA fragments (RPFs) was approximately 28 nt (***Figure 3—figure supplement 9A***). The RPFs exhibited a significant 3-nt periodicity. Metagene analysis of individual 28 nt reads revealed the distribution of RPFs across the gene locus in *w$^{1118}$* and *Mettl5$^{1bp}$*, respectively. The starting point of translation is 12 nt upstream of the start codon and gradually disappears 15 nt from the stop codon (***Figure 3—figure supplement 7G–L***). RPFs on the metagene plot distribution around the translation start and translation stop site (***Figure 3—figure supplement 8A–F***), different coding frames on CDS, 3'-UTR and 5'-UTR, respectively, in two groups of samples all showed the periodicity (***Figure 3—figure supplement 8G–L***). As expected, all plots show an enrichment of P-sites in the first frame on the coding sequence but not the UTRs, in accord with ribosome protected fragments from protein coding mRNAs (***Figure 3—figure supplement 8G–L***).

Analysis of open reading frame (ORF) types revealed alterations in their distribution. Compared to $w^{1118}$, $Mettl5^{1bp}$ exhibited an increased proportion of overlapping downstream ORFs (dORFs) and fewer non-overlapping dORFs (*Figure 3—figure supplement 9B*). In both groups, translated upstream ORFs (uORFs) were significantly shorter than untranslated uORFs ($p < 0.05$) (*Figure 3—figure supplement 9C*). Additionally, $Mettl5^{1bp}$ displayed reduced read counts for both dORFs and uORFs (*Figure 3—figure supplement 9D, E*). Based on translation potential, uORFs were classified as either translated or untranslated, and motif analysis was performed separately for each category (*Figure 3—figure supplement 9F, G*).

## Mutation of *Mettl5* altered codon preference

Comparison of codon occupancy (A-site) between the two groups revealed that $Mettl5^{1bp}$ preferentially used GAC and GAU, whereas $w^{1118}$ favored UCC (*Figure 3—figure supplement 9H*). This trend was further supported by the cumulative frequency distribution of these codons (*Figure 3—figure supplement 9I–K*). Since GAC and GAU both encode aspartate (Asp), we analyzed Asp amino acid occupancy. Intriguingly, Asp was significantly enriched in $w^{1118}$ during translation (*Figure 3—figure supplement 9L*), suggesting that $Mettl5^{1bp}$ may exhibit altered translational regulation (see Discussion for details).

Metagene analysis of RPFs revealed distinct translation patterns between $Mettl5^{1bp}$ and $w^{1118}$. The coding sequence (CDS) and flanking regions were segmented into 100 equal bins, and average RPF density was computed for each bin. The resulting plots illustrate differences in ribosome occupancy between $Mettl5^{1bp}$ and $w^{1118}$ along the CDS (*Figure 3—figure supplement 9M*), near the translation start site (*Figure 3—figure supplement 9N*) and around the translation termination site (*Figure 3—figure supplement 9O*). Notably, $Mettl5^{1bp}$ and $w^{1118}$ exhibited divergent ribosome occupancy patterns, particularly along the CDS and near the start codon (*Figure 3—figure supplement 9M, N*), suggesting potential differences in translation dynamics and initiation efficiency.

## *Mettl5* regulates the clock gene regulatory loop

Our findings demonstrate that $Mettl5^{1bp}$ disrupts the core clock gene regulatory loop controlling circadian rhythm. We observed significant alterations in both transcriptional and translational levels of multiple clock genes, with *cry* and *Clk* showing upregulation while *tim*, *per*, *vri*, and *pdp1* were downregulated at both levels (*Figure 4A, B*). Notably, *per*, *vri*, and *pdp1* exhibited particularly pronounced downregulation in translation efficiency (*Figure 4C*). These changes occurred without affecting clock neuron morphology at different time points (*Figure 4—figure supplement 1*).

The observed expression patterns revealed an unexpected regulatory relationship: while the canonical clock circuitry positions *per* downstream of *Clk*, our finding that *Clk* was upregulated while *per* was downregulated suggests *per* may actually function upstream of *Clk* in *Mettl5*-mediated regulation. Surprisingly, despite the transcriptional downregulation of *per* (*Figure 4D*), we detected increased PER protein levels through both immunostaining and western blot analyses (*Figure 4E–H*). Detection of the PER protein at different time points indicated that it was increased at both ZT0 and ZT18 (*Figure 4E, F*, *Figure 4—source data 1*, *Figure 4—source data 2*). This apparent contradiction aligns with the observed circadian phenotype, as *Mettl5* mutants showed significantly longer period lengths (*Table 1*), mirroring effects seen when PER stabilization results from reduced kinase activity as previously reported (*Philpott et al., 2023*). Genetic epistasis experiments further supported this model, with clock gene mutants modifying *Mettl5* mutant phenotypes that suggest both *Clk* and *per* downstream of *Mettl5* (*Figure 4I–N*, *Table 1*). Secondary effects may exist for the significant increase in daytime sleep in the double mutants. Together, these results indicate that $Mettl5^{1bp}$ affects circadian regulation through mechanisms that extend beyond transcriptional control, likely involving post-translational regulation of PER protein stability.

To investigate the factors mediating PER protein level changes in *Mettl5* mutants, we examined the ubiquitin–proteasome pathway, which plays a well-documented role in Period protein degradation (*Grima et al., 2002*; *Ko et al., 2002*; *Chiu et al., 2008*). Notably, recent evidence indicates that $m^6A$ regulates the ubiquitin–proteasome system in other biological contexts (*Sun et al., 2023*). Our integrated analysis of RNA-seq and Ribo-seq data revealed significant downregulation of multiple proteasome pathway components in $Mettl5^{1bp}$ mutants at both transcriptional and translational levels (*Figures 3E, I and 4O*), suggesting impaired protein degradation capacity. Based on these findings,

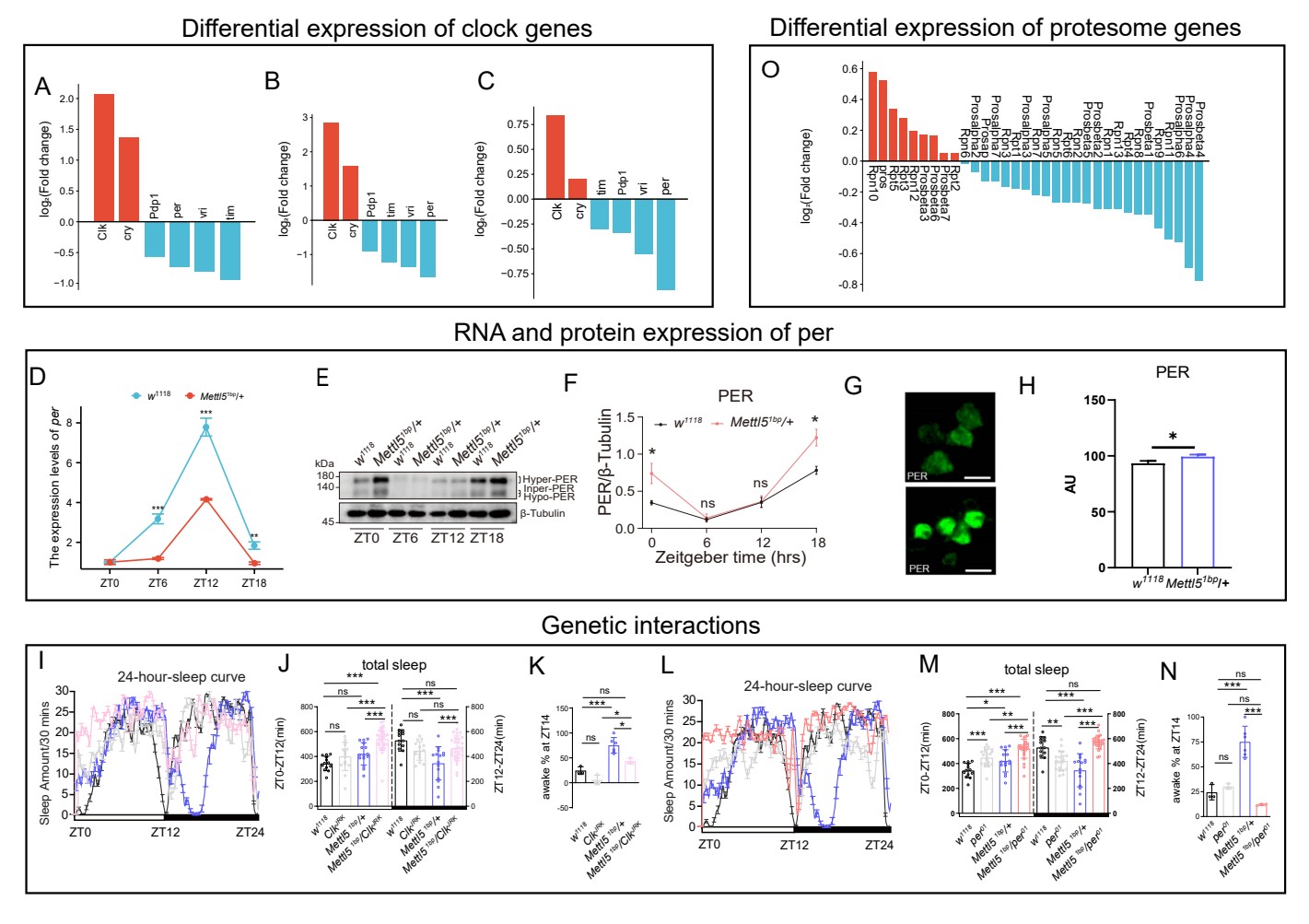

**Figure 4.** Clock genes expression mediated the sleep phenotype caused by *Mettl5* mutation. (**A–C**) Fold changes in clock genes with significant expression level differences between $w^{1118}$ and *Mettl5*$^{1bp}$ were observed in RNA-seq, Ribo-seq, and translation efficiency analyses. (**D**) The gene expression levels of *per* at four different time points of $w^{1118}$ and *Mettl5*$^{1bp}$. (**E**) Representative western blot analysis of PER protein levels in w1118 and Mettl51bp/+fly heads collected at four distinct time points (ZT0, ZT6, ZT12, and ZT18). Brackets indicate different phosphorylation states of PER: hyper-phosphorylated (Hyper), intermediate (Inper), and hypo-phosphorylated (Hypo). β-Tubulin was used as a loading control. (**F**) Quantification of total PER protein levels relative to β-tubulin. Data are presented as mean ± SEM from three independent biological replicates ($n$ = 3). Statistical significance was determined by unpaired Student's *t*-test at each time point. * stands for p < 0.05; ns stands for not significant. (**G**) Representative image of PER protein immunofluorescence staining at ZT0 in the small ventral lateral neurons (small LNvs). (**H**) Statistical analysis of the immunofluorescence intensity for PER in small LNvs. (**I**) Sleep curve throughout the day for *Mettl5*$^{1bp}$, *per*$^{01}$, double mutant and control flies. (**J**) Total sleep for *Mettl5*$^{1bp}$, *per*$^{01}$, double mutant and control flies. (**K**) Percentage of awake time in *Mettl5*$^{1bp}$ flies, partially rescued by double mutant flies. (**L**) Sleep curve throughout the day for *Mettl5*$^{1bp}$, *per*$^{01}$, double mutant, and control flies. (**M**) Total sleep for *Mettl5*$^{1bp}$, *per*$^{01}$, double mutant, and control flies. (**N**) Percentage of awake time in *Mettl5*$^{1bp}$ flies, partially rescued by double mutant flies. (**O**) Fold changes in proteasome subunits with significant expression level differences between $w^{1118}$ and *Mettl5*$^{1bp}$ were observed in RNA-seq, Ribo-seq, and translation efficiency analyses. For statistical significance, * stands for p < 0.05, ** stands for p < 0.01, *** stands for p < 0.001, ns stands for not significant.

The online version of this article includes the following source data and figure supplement(s) for figure 4:

**Source data 1.** PDF file containing original western blots for **Figure 4E**, indicating the relevant bands, phosphorylation states, and genotypes.

**Source data 2.** Original files for western blot analysis displayed in **Figure 4E**.

**Figure supplement 1.** Morphology of clock neurons in *Mettl5* mutant.

**Table 1.** Circadian rhythm phenotypes of various mutants.

| Genotype | NumTotal | %Rhythmic | Period | Signif vs $w^{1118}$ (Period) | Power | Signif vs $w^{1118}$ (Power) |
|---|---|---|---|---|---|---|
| $w^{1118}$ | 32 | 92.6 | 23.9 ± 0.05 | | 127.9 ± 7.62 | |
| $Mettl5^{1bp}$/+ | 32 | 92.3 | 28.3 ± 0.4 | *** | 114.3 ± 6.76 | ns |
| $Mettl5^{1bp}$/+; UAS-$Mettl5$/$Mettl5$-Gal4 | 32 | 96.9 | 24 ± 0.02 | ns | 127.1 ± 5.42 | ns |
| $Mettl5^{1bp}$/$Clk^{JRK}$ | 44 | 29.5 | 24 ± 0.04 | ns | 68.5 ± 8.85 | *** |
| $Mettl5^{1bp}$/$Per^{01}$ | 54 | 9.3 | 24.2 ± 0.2 | ns | 43.8 ± 13.41 | *** |

we propose a model where $Mettl5$ regulates circadian function through three interconnected mechanisms: first, by directly modulating proteasome components to control PER protein stability post-translationally; second, by transcriptionally and translationally regulating $per$ and other $Clk$ expression. In this model, $Mettl5^{1bp}$-induced proteasome downregulation leads to PER accumulation, which is responsible for the phenotypes (**Figure 5**).

### $Mettl5^{1bp}$ alters axon complexity

Previous studies have established a strong correlation between sleep homeostasis and synaptic complexity (**Bushey et al., 2011**). Additionally, synaptogenesis has been shown to enhance proteasome activity in axons (**Costa et al., 2019**). Given our observations of impaired sleep rebound following deprivation and altered expression of proteasome subunits in $Mettl5^{1bp}$ mutants, we sought to examine potential effects on synaptic complexity.

To assess synaptic complexity, we adapted an established quantification method (**Bushey et al., 2011**) using $Syt$-GFP, a marker that colocalizes with endogenous synaptic vesicles, to visualize presynaptic morphology changes. Control experiments with UAS-$Fmr1$ and $Fmr1$ mutations successfully replicated the expected decrease and increase in $syt$-eGFP signal, respectively (**Figure 6A–D**). Strikingly, $Mettl5^{1bp}$ mutants exhibited significantly increased $syt$-eGFP fluorescence in presynaptic terminals (**Figure 6E–M**), indicating altered synaptic complexity.

## Discussion

Our study reveals that $Mettl5$, a known rRNA methyltransferase, modulates sleep through its RNA methylation activity. Through integrated RNA-seq and Ribo-seq analyses of $Mettl5^{1bp}$ mutants, we identified $Mettl5$'s downstream targets at both transcriptional and translational levels. Further investigation demonstrated that $Mettl5$ influences sleep regulation by affecting two key pathways: the circadian clock gene network and the proteasome system. These findings provide novel mechanistic insights into sleep control, highlighting the coordinated role of protein synthesis and degradation in this process. Notably, our Ribo-seq analysis revealed that $Mettl5^{1bp}$ alters fundamental translation features, including uORF translation efficiency and codon preference, suggesting rRNA methylation plays a regulatory role in these processes.

This discovery has important clinical implications, as $METTL5$, the human ortholog of $Mettl5$,

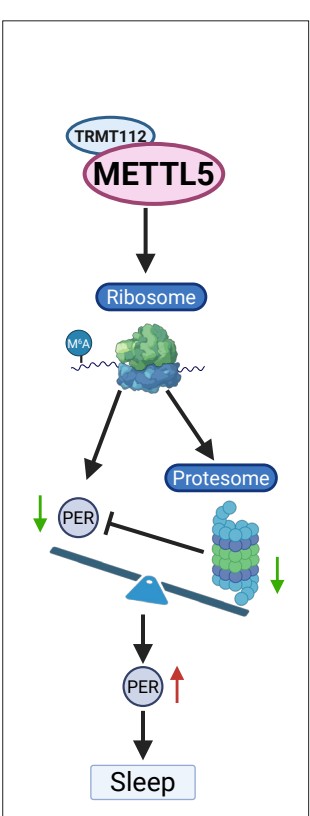

**Figure 5.** A working model illustrating the role of $Mettl5$ in $Drosophila$ sleep was presented.

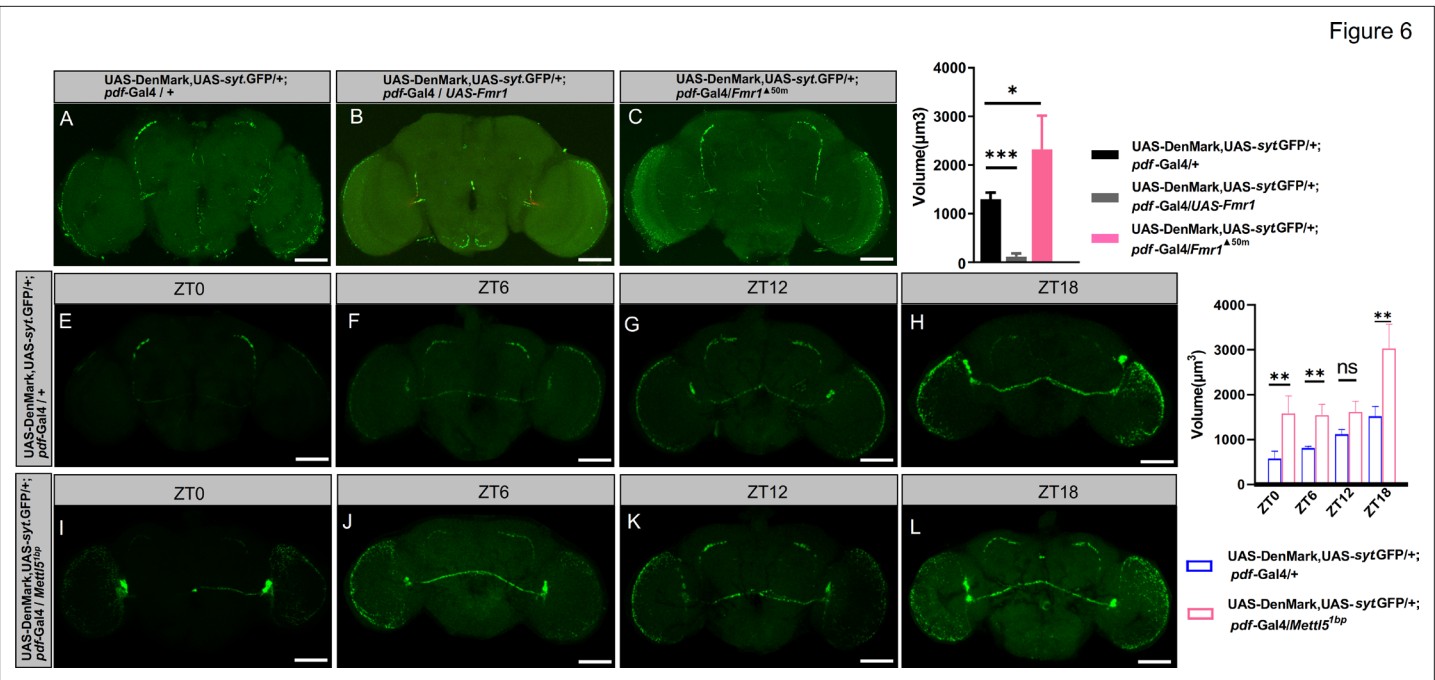

**Figure 6.** The axon complexity was found to be affected by *Mettl5¹ᵇᵖ*. (**A–C**) Representative confocal micrographs of small ventral lateral neuron (s-LNv) axonal terminals across different genotypes. Presynaptic structures were visualized using *syt*-eGFP (green), which colocalizes with endogenous synaptic vesicles: (**A**) control, (**B**) *Fmr1* overexpression, and (**C**) *Fmr1* null mutant. (**D**) Quantification of s-LNv axonal terminal volumes corresponding to genotypes in **A–C**. (**E–H**) Representative images of axonal terminal morphology in control flies at four time points: ZT0, ZT6, ZT12, and ZT18. (**I–L**) Representative images of axonal terminal morphology in *Mettl5* mutant flies at ZT0, ZT6, ZT12, and ZT18. (**M**) Quantitative comparison of axonal terminal volumes between control and *Mettl5* mutant flies at different time points. Scale bar: 50 μm. For statistical significance, * stands for p < 0.05, ** stands for p < 0.01, *** stands for p < 0.001, ns stands for not significant.

is associated with ID when mutated (*Richard et al., 2019*). Our work expands the understanding of *Mettl5*'s molecular function and may inform potential therapeutic strategies for ID. The clinical relevance of our findings is underscored by reports of sleep disturbances, particularly reduced sleep duration, in ID patients—a phenotype that parallels our observations in *Mettl5¹ᵇᵖ* mutants. The mechanistic framework established in this study could explain these clinical sleep abnormalities. However, further validation in vertebrate models is needed to determine whether this regulatory mechanism is evolutionarily conserved and applicable to human sleep disorders.

As shown in *Table 1*, the *Mettl5¹ᵇᵖ*/+ mutant exhibits a robust long-period phenotype, with circadian rhythms significantly extended to 28.3 ± 0.4 hr compared to the wild-type's 23.9 ± 0.05 hr. This prolonged period perfectly aligns with the observed behavioral phenotypes, including delayed nighttime sleep onset, later daytime waking, and the overall shift in sleep profile. This is indeed quite similar to a previous report on *PERIOD3* variant (*Zhang et al., 2016*). We think that the prolonged circadian period contributes to the observed sleep phenotype. However, since total sleep time was significantly reduced in the mutant, we cannot attribute the phenotype solely to period lengthening. Furthermore, our 24-hr PER expression analysis in *Mettl5* mutants revealed elevated PER protein levels at ZT1 and ZT18, while ZT6 and ZT12 showed no significant changes, with no apparent phase shift. These findings collectively suggest that the phenotype primarily results from PER protein stabilization and accumulation.

We found that *Mettl5* heterozygotes showed significant reductions in total RNA and 18S rRNA methylation levels, contrasting with mouse studies where heterozygous knockouts maintained normal 18S rRNA m6A methylation (*Sepich-Poore et al., 2022*). This discrepancy may stem from either a fundamental difference in rRNA methylation regulation between *Drosophila* and mice, or distinct biological consequences of knockdown versus knockout approaches, as complete gene elimination often triggers compensatory mechanisms (*Teng et al., 2013*; *Rossi et al., 2015*; *Vu et al., 2015*; *Ma et al., 2019*; *El-Brolosy et al., 2019*).

Our study uncovers a previously unrecognized connection between circadian clock genes and proteasome function. While previous work demonstrated that the circadian clock rhythmically regulates proteasome components in *Drosophila* fat bodies under dietary restriction (*Hwangbo et al., 2023*), we now show that *Mettl5* modulates clock protein synthesis and degradation in clock neurons by influencing proteasome activity. This regulation likely occurs through *Mettl5*-mediated ribosomal methylation in clock neurons, which impacts the proteasome degradation pathway (*Costa et al., 2019*), ultimately altering clock protein dynamics.

*Mettl5* represents a novel integrator of rRNA methylation and proteasome function, providing a mechanism to balance protein synthesis and degradation. Interestingly, we observed specific effects on both transcriptional and translational outputs, with particular proteasome subunits showing differential regulation. This specificity may arise from either selective translational control by *Mettl5* or additional layers of regulation through protein–protein interactions. These findings suggest an intricate regulatory network coordinating these processes, though further studies are needed to elucidate the underlying mechanisms.

Our results reveal complex relationships between clock genes and sleep regulation. While *cyc* loss-of-function alleles show enhanced sleep rebound in females (*Shaw et al., 2002*), *Mettl5*[1bp] mutants with elevated PER protein exhibit reduced rebound. Notably, *Mettl5*'s regulatory effects display tissue specificity—although absent in canonical clock neurons, *Mettl5*-Gal4 is expressed in distinct neurons and glia that appear crucial for sleep regulation. This expression pattern, combined with Clock's broader distribution (*Patop et al., 2023*), suggests specialized circuits for sleep homeostasis that warrant further investigation.

Evidence indicates that sleep functions in development, metabolism, and neuronal plasticity (*Anafi et al., 2019*). Sleep exerts effects on neuronal plasticity by modifying synapses. The synaptic homeostasis theory proposes that sleep has a role in downscaling synaptic strength (*Tononi and Cirelli, 2006*). Indeed, synapse markers progressively decrease during sleep in both mammals and *Drosophila* brains (*López-Varea et al., 2021*; *Vyazovskiy et al., 2008*; *Gilestro et al., 2009*; *Liu et al., 2010*). More importantly, evidence indicates that sleep need and protein levels in synapses are tightly linked. Studies of the presynaptic active zone have shown that synaptic plasticity regulates sleep homeostasis (*Huang et al., 2020*). Genome-wide proteomic studies of synapses in mouse brains indicate that the synaptic proteins peak around dusk and dawn (*Noya et al., 2019*; *Brüning et al., 2019*). Sleep deprivation experiments have demonstrated that the sleep drive is significantly dependent on the cycling of proteins and phosphoproteins in synapses, in contrast to mRNAs (*Noya et al., 2019*; *Brüning et al., 2019*). These results suggest that gene expression regulation at the protein level is crucial for sleep. Consistently, we detected changes in the quantity of axons in *Mettl5*[1bp], suggesting a possible regulation of neural circuits by *Mettl5*.

The mechanism by which *METTL5* regulates translation warrants further investigation. Previous studies have demonstrated that *METTL5* influences translation (*Rong et al., 2020*; *Peng et al., 2022*), but whether the mechanisms identified here are conserved across other systems remains an intriguing question. In our analysis, we observed increased usage of aspartate (Asp) codons in *Mettl5* mutants. Notably, prior work has linked codon usage to PER protein function—specifically, a codon-optimized version of PER failed to rescue circadian rhythmicity in *per* mutant flies, unlike the wild-type version (*Fu et al., 2016*). Further analysis revealed that PER protein levels were elevated in these mutants, suggesting that codon optimization enhances PER expression (Figure 2B in *Fu et al., 2016*). Strikingly, when we examined the codon-optimized region from *Fu et al., 2016*, we found that GAC (Asp) was highly enriched, raising the possibility that *Mettl5* mutation affects PER protein accumulation by altering GAC codon usage. Additional experiments will be needed to validate this hypothesis. Furthermore, we detected changes in uORFs in *Mettl5* mutants, but their relationship to translational regulation requires further exploration.

Our study demonstrates that ribosomal components can exert gene-specific regulatory functions, building upon previous work showing context-dependent ribosome specialization (*Simsek et al., 2017*). In *Mettl5* mutants, we observed distinct alterations in the translational efficiency of specific genes. These effects could potentially arise through two non-exclusive mechanisms: (1) *Mettl5*-mediated rRNA modifications may modulate ribosomal binding affinity for particular mRNA sequences, or (2) these modifications might contribute to the formation of specialized ribosome populations that preferentially translate specific subsets of mRNAs. Future studies will be required to distinguish between these possibilities and fully elucidate the underlying molecular mechanisms.

## Materials and methods

### *Drosophila* strains

Fly stocks used in this study were maintained under standard culture conditions. We used the $w^{1118}$ as the control strain. The following flies were obtained from Bloomington Stock Center: $w^{1118}$ (Bl: 5905), *Mettl5*-Gal4 (Bl: 19514), *Repo*-Gal4 (Bl: 7415), *nSyb*-Gal4 (Bl: 51941), *pdf*-Gal4 (Bl: 41286), $Clk^{JRK}$ (Bl: 80927), $per^{01}$ (Bl: 80917), $Fmr1^{\Delta 50M}$ (Bl: 6930), UAS-*Trmt112*RNAi (VDRC: 101515), UAS-DenMark, UAS-*syt*.eGFP (Bl: 33064), UAS-GFPstinger (from Yi Rao's lab), UAS-*Fmr1* (Bl: 6931), and UAS-*Mettl5* (FlyORF: F000760) are from Fly ORF collection. UAS-*Mettl5m*-3HA (generated in this study).

*Mettl5*$^{1bp}$ and *Mettl5*$^{9bp}$ mutants were generated by the CRISPR/Cas system as described previously (*Cheng et al., 2020*). The target gRNA was designed with an online tool: http://tools.flycrispr.molbio. wisc.edu/targetFinder/. A target sequence was chosen that has the sequence of 5'- GAGTGGGG TACTGTTCCAAC**AGG** with the PAM sequence in bold. Mutations were verified by genomic PCR and sequencing.

### Sleep behavior assays

All sleep assays were conducted in a controlled environment incubator maintained at 25 ± 1°C with 60% ± 5% relative humidity. We maintained a 12:12 light-dark cycle with lights on at ZT0 (06:30) and off at ZT12 (18:30). Fly activity was monitored using the *Drosophila* Activity Monitoring (DAM) System (Trikinetics, Waltham, MA). Following a 2-day acclimation period, we recorded locomotor activity at 1 min intervals for 3 consecutive days. Data analysis was performed using pysolo (*Gilestro and Cirelli, 2009*), with sleep defined as ≥5 min of continuous inactivity.

Mechanical sleep deprivation was performed using the SNAP method to keep flies awake for 12 hr overnight (*Shaw et al., 2002*). Sleep deprivation is applied by mechanical stimuli using a timer-controlled rotating shaker. The intensity of stimuli was 1500 rpm once every minute (2 s/1 min) (*Shimizu et al., 2008*). Baseline sleep was established from 24-hr recordings prior to deprivation. Sleep loss and recovery were quantified according to previous publication (*Cirelli et al., 2005*).

Regarding the 'awake %' metric, it indicates that at specific time points (e.g., ZT14), the percentage of awake fruit fly population at that moment. At ZT19, we evaluated arousal thresholds by administering a standardized gentle mechanical stimulus. Responsiveness was determined by monitoring activity for 1-min post-stimulation. Flies showing no activity during this window were scored as non-responsive. We calculated arousal percentages from the proportion of flies that awakened in response to stimulation.

Circadian rhythm of individual male flies was measured using the DAM System (Trikinetics). Male flies were loaded individually into glass tubes with a length of 65 mm and an inner diameter of 5 mm. The tubes contained standard cornmeal fly food at one end and were sealed with a cotton stopper at the other end. The flies were entrained to a 12-hr light/12-hr dark cycle for 3 days and then released to constant darkness for at least 6 days to measure their rhythmicity. Data analysis is done on a Macintosh computer running the FaasX (Fly activity analysis suite) software.

### Statistical analysis

All statistical analyses were performed using GraphPad Prism 5 software. For sleep parameter comparisons, we used nonparametric tests including the two-tailed Mann–Whitney test for pairwise comparisons and one-way ANOVA with Tukey's post hoc test for multiple comparisons. qPCR data were analyzed using unpaired Student's $t$-tests. The specific statistical test used for each experiment is indicated in the corresponding figure. In all analyses, a p-value of less than 0.05 was considered statistically significant.

### Quantitative PCR

Total RNA was extracted from cells and tissues using the TRNzol Universal Reagent (Tiangen #DP4-02). For cDNA synthesis, we employed the PrimeScript RT reagent Kit with gDNA Eraser (TAKARA #RR047A) following the manufacturer's protocol. Quantitative PCR was carried out using SuperReal PreMix Plus (SYBR Green) (Tiangen #DP4-02). *RP49* served as the endogenous reference gene for normalization across samples. All experiments included three independent biological replicates to ensure reproducibility. The primers used in this experiment are *RP49*-F: CGGTTACGGATCGAAC

AAGC; *RP49*-R: CTTGCGCTTCTTGGAGGAGA; *Mettl5*-F: CGGTTTCTGGAGGTGGC; *Mettl5*-R: GCTGGCGTCGATGTTGTAC.

## Imaging and analysis of axon volume

We quantified small LNvs axon volume using ImageJ's Object Counter 3D plugin to measure pixel counts from raw images. Image processing involved applying a standardized threshold that clearly visualized intact axons in control samples. For consistent measurements, we specifically analyzed the axon span between the first axonal bifurcation and the terminal tip, as indicated by the white rectangular markers in *Figure 6*.

## Sample collection and library construction for Ribo-seq and RNAseq

For sample collection, $w^{1118}$ and $Mettl5^{1bp}$ *Drosophila* were harvested at ZT15, immediately transferred to centrifuge tubes, and flash-frozen in liquid nitrogen. Fly heads were subsequently separated and collected while frozen, with three biological replicates prepared for each genotype. Each sample was equally divided for parallel RNA-seq and Ribo-seq analyses. For Ribo-seq library preparation, samples were lysed in buffer containing 50 mg/ml cycloheximide (Novogen, China) to preserve ribosome positioning, followed by RNase I digestion to generate RPFs. Monosomes were isolated using MicroSpin S-400 HR size-exclusion chromatography, followed by rRNA depletion and PAGE purification to select 20–38 nt RPFs. Purified fragments underwent end repair, adapter ligation, reverse transcription, and PCR amplification before Illumina PE150 sequencing. For RNA-seq, total RNA was extracted using TRIzol reagent, with cDNA libraries prepared and sequenced using Illumina PE150 by Novogen. Raw sequencing data in FASTQ format were processed to remove adapter sequences, reads containing N bases, and low-quality reads, while simultaneously calculating Q20/Q30 scores and GC content to generate clean reads for downstream analysis.

## RNA-seq analysis

We performed genome alignment and transcriptome analysis using the following pipeline: First, we built a Hisat2 index (v2.0.5) for the *Drosophila melanogaster* reference genome (dm6 assembly). Clean paired-end reads were then aligned to this reference using Hisat2 (v2.0.5) (*Kim et al., 2019*). The resulting alignments were processed using StringTie (v1.3.3b) (*Pertea et al., 2015*) for reference-based transcript assembly. For gene-level quantification, we used FeatureCounts (v1.5.0) (*Liao et al., 2014*) to count reads mapped to each annotated gene. Differential expression analysis between $w^{1118}$ and $Mettl5^{1bp}$ was conducted with DESeq2 (v1.20.0) (*Pertea et al., 2015*), with genes meeting both criteria (p.adjust <0.05 and absolute $\log_2$ (fold change) ≥1) considered statistically significant.

## Ribo-seq analysis

We implemented a comprehensive Ribo-seq analysis pipeline beginning with quality filtering using Bowtie (*Liao et al., 2014*) to remove reads aligning to non-coding RNAs (rRNA, tRNA, snoRNA, snRNA from FlyBase Release 6.13) with a 2-mismatch allowance (-v 2). The remaining RPFs were mapped to the *Drosophila melanogaster* genome (FlyBase Release 6.13) using STAR (v2.7.3a) (*Dobin et al., 2013*), followed by transcript-level alignment to protein-coding sequences using Bowtie (v1.2.2) (*Langmead et al., 2009*) with parameters '-a -v 2'. CDS-aligned RPFs were quantified using feature-Counts (Subread v1.6.3) and normalized as RPKM. Differential expression analysis was performed with DESeq2 (v1.14.1) (*Love et al., 2014*) using thresholds of $|\log_2 FC|$ ≥0.265 and p.adj <0.05, while translation efficiency differences were assessed using RiboDiff. Data quality was verified through riboWaltz (v1.1.0) (*Lauria et al., 2018*) for 3-nt periodicity and reading frame analysis, with Ribocode (*Xiao et al., 2018*) employed for P-site positioning and uORF motif analysis. Functional enrichment analyses (GO, KEGG, and GSEA) were conducted using clusterProfiler (v4.5.2.002) (*Wu et al., 2021*), with results visualized through ggplot2-generated plots and GO term simplification performed using SimplifyEnrichment (*Gu and Hübschmann, 2023*).

## Immunofluorescence experiments

We performed immunofluorescence staining on 7- to 15-day-old adult flies (unless otherwise specified). Flies were anesthetized with $CO_2$ and dissected in ice-cold 0.03% PBST (1× PBS with 0.03% Triton X-100; Sigma, T9284). Samples were fixed in 2% paraformaldehyde for 55 min at room temperature

(RT), followed by four 15 min washes in 0.03% PBST at RT. After blocking overnight at 4°C in 10% Normal Goat Serum (NGS; in 1× PBS with 2% Triton), samples were incubated with primary antibodies for 24 hr at 4°C. Primary antibodies included rat anti-Elav (DSHB, 9F8A9; 1:200) and mouse anti-Repo (DSHB, 8D12; 1:200), diluted in antibody buffer (1.25% PBST, 1% NGS). Following four 15 min washes in 1× PBS with 1% Triton at RT, samples were incubated overnight at 4°C with secondary antibodies: Alexa Fluor 568 (Thermo Fisher, A11004; 1:200) and Alexa Fluor 647 (Thermo Fisher, A21247; 1:200). After four additional 15 min washes in 1× PBS with 1% Triton at RT, samples were mounted using DAPI-containing antifade mounting medium (Solarbio, S2110).

Images were acquired using a Leica SP8 confocal microscope with LAS X software, applying auto Z-brightness correction when needed for signal uniformity. Images were processed in Adobe Photoshop CS6 and figures assembled in Adobe Illustrator 2020. PDF immunofluorescence followed the same protocol. All experiments included ≥3 biological replicates, each containing ≥10 flies. Fluorescence intensity was quantified using ImageJ.

## Western blotting

Protein samples were prepared by homogenizing approximately 30 fly heads in RIPA lysis buffer (150 mM NaCl, 1.0% NP-40, 0.5% sodium deoxycholate, 0.1% SDS, 50 mM Tris-HCl, pH 8.0) supplemented with protease (CW2200S) and phosphatase (CW2383S) inhibitor cocktails according to manufacturer specifications. Lysates were mixed with 2× SDS loading buffer, boiled at 100°C for 5 min, and immediately cooled on ice.

For immunoblotting, membranes were probed with rabbit anti-PER primary antibody (1:5000 dilution; kindly provided by Dr. Jeffrey Price's laboratory, University of Missouri-Kansas City) overnight at 4°C, followed by incubation with HRP-conjugated goat anti-rabbit IgG secondary antibody (1:1500; ABclonal, AS014) for 4 hr at RT. Protein signals were detected using ECL substrate (ABclonal, RM00021P) and imaged with an Amersham ImageQuant 800 system (GE Healthcare). Band intensities were quantified using ImageJ software, with three biological replicates performed for statistical reliability.

## LC–MS/MS analysis of $m^6A$ levels

Total RNA was isolated from cells and tissues using TRNzol Universal Reagent (Tiangen, #DP4-02). For 18S rRNA purification, we separated total RNA by polyacrylamide gel electrophoresis followed by gel extraction. Our analysis utilized ribonucleoside standards including adenosine (rA) and $N^6$-methyladenosine ($N^6$mA), with a mobile phase consisting of methanol: $ddH_2O$ (vol/vol).

For each biological replicate, 1 µg of total RNA or purified 18S rRNA was digested to single nucleosides using Nucleoside Digestion Mix (NEB, #M0649). Proteins were precipitated by adding a 4:1 methanol: digest ratio and incubating at –20°C for 2 hr. Quantitative analysis was performed using multiple reaction monitoring with the following transitions: 268.10275 → 136.0621 (rA), 282.11835 → 150.0774 ($N^6$mA).

## Negative geotaxis RING assay

Climbing assays were performed following a 12-hr recovery period after $CO_2$ anesthesia, using 200 flies per genotype distributed across 10 vials (20 flies/vial) marked at 90 mm height. Each assay consisted of three trials separated by 15-min intervals, initiated by sharply tapping vials three times to induce negative geotaxis, with the number of flies reaching the 90 mm mark within 10 s recorded per trial. Five groups were tested per genotype (15 trials total). All assays were video recorded under standardized conditions with the camera positioned 20 cm from vials and uniform backlighting provided by a white open-faced box to ensure consistent imaging quality.

## Acknowledgements

This work was supported by National Natural Science Foundation of China (Grant Nos. 32070492 and 32122017) to Juan Du. We would also like to acknowledge the 2115 Talent Development Program of China Agricultural University.

## Additional information

### Funding

| Funder | Grant reference number | Author |
| --- | --- | --- |
| National Natural Science Foundation of China | 32070492 | Juan Du |
| National Natural Science Foundation of China | 32122017 | Juan Du |

The funders had no role in study design, data collection, and interpretation, or the decision to submit the work for publication.

### Author contributions

Xiaoyu Wu, Formal analysis, Investigation, Methodology, Writing – original draft; Xingzhuo Yang, Data curation, Software, Formal analysis, Investigation, Methodology, Writing – original draft; Tiantian Fu, Investigation; Yikang Rong, Resources, Writing – original draft; Juan Du, Conceptualization, Resources, Formal analysis, Funding acquisition, Writing – original draft, Project administration, Writing – review and editing

### Author ORCIDs

Xiaoyu Wu (ID) https://orcid.org/0009-0001-7183-899X
Xingzhuo Yang (ID) https://orcid.org/0009-0000-0025-5503
Juan Du (ID) https://orcid.org/0000-0002-1850-3613

Reviewer #1 (Public review): https://doi.org/10.7554/eLife.103427.4.sa1
Reviewer #3 (Public review): https://doi.org/10.7554/eLife.103427.4.sa2
Author response https://doi.org/10.7554/eLife.103427.4.sa3

## Additional files

### Supplementary files

MDAR checklist

### Data availability

The RNA-seq and Ribo-seq data from this publication have been deposited to the NCBI bioproject database https://www.ncbi.nlm.nih.gov/bioproject/ and assigned the identifier PRJNA994860.

The following dataset was generated:

| Author(s) | Year | Dataset title | Dataset URL | Database and Identifier |
| --- | --- | --- | --- | --- |
| Yang X | 2024 | RNA-seq and Ribo-seq revealed the downstream events in Mettl5 mutation | https://www.ncbi.nlm.nih.gov/bioproject/?term=PRJNA994860 | NCBI BioProject, PRJNA994860 |

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
