## [Editor Report · eLife Assessment]

The authors present **useful** findings demonstrating that the RNA modification enzyme Mettl5 regulates sleep in *Drosophila*. Through transcriptome- and proteome-wide analyses, the authors identified downstream targets affected in heterozygous mutants and proposed that Mettl5 regulates the translation and degradation of clock genes to maintain normal sleep function. Through additional analyses, the authors provided **solid** evidence supporting this model.

---

## [Referee Report · Reviewer #1 (Public review)]

Summary:

Here the authors attempted to test whether the function of Mettl5 in sleep regulation was conserved in *Drosophila*, and if so, by which molecular mechanisms. To do so they performed sleep analysis, as well as RNA-seq and ribo-seq in order to identify the downstream targets. They found that the loss of one copy of Mettl5 affects sleep, and that its catalytic activity is important for this function. Transcriptional and proteomic analyses show that multiple pathways were altered, including the clock signaling pathway and the proteasome. Based on these changes the authors propose that Mettl5 modulate sleep through regulation of the clock genes, both at the level of their production and degradation, possibly by altering the usage of Aspartate codon.

Comments on revisions:

The authors addressed all my comments satisfactorily.

---

## [Referee Report · Reviewer #3 (Public review)]

Xiaoyu Wu and colleagues examined a potential role in sleep of a *Drosophila* ribosomal RNA methyltransferase, mettl5. Based on sleep defects reported in CRISPR generated mutants, the authors performed both RNA-seq and Ribo-seq analyses of head tissue from mutants and compared to control animals collected at the same time point. A major conclusion was that the mutant showed altered expression of circadian clock genes, and that the altered expression of the period gene in particular accounted for the sleep defect reported in the mettl5 mutant. In this revision, the authors have added a more thorough analysis of clock gene expression and show that PER protein levels are increased relative to wild type animals a specific times of day, indicating increased stability of the protein. Given that PER inhibits its own transcription, the per RNA is low in the mutants. The revised manuscript included efforts toward a more detailed understanding of how clock gene expression was altered in the mutants, as well as other clarification of sleep phenotypes.

Comments on revisions:

All critiques have been addressed by the authors; the manuscript is much improved from its original submission. Thank you.

---

## [Author Response]

The following is the authors’ response to the previous reviews

**Public Reviews:**

**Reviewer #1 (Public review):**
Here, the authors attempted to test whether the function of Mettl5 in sleep regulation was conserved in *Drosophila*, and if so, by which molecular mechanisms. To do so they performed sleep analysis, as well as RNA-seq and ribo-seq in order to identify the downstream targets. They found that the loss of one copy of Mettl5 affects sleep, and that its catalytic activity is important for this function. Transcriptional and proteomic analyses show that multiple pathways were altered, including the clock signaling pathway and the proteasome. Based on these changes the authors propose that Mettl5 modulate sleep through regulation of the clock genes, both at the level of their production and degradation, possibly by altering the usage of Aspartate codon.Comments on revised version:The authors satisfactorily addressed my comments, even though the precise mechanism by which Mettl5 regulates translation of clock genes remains to be firmly demonstrated.
**Reviewer #3 (Public review):**
Xiaoyu Wu and colleagues examined a potential role in sleep of a *Drosophila* ribosomal RNA methyltransferase, mettl5. Based on sleep defects reported in CRISPR generated mutants, the authors performed both RNA-seq and Ribo-seq analyses of head tissue from mutants and compared to control animals collected at the same time point. A major conclusion was that the mutant showed altered expression of circadian clock genes, and that the altered expression of the period gene in particular accounted for the sleep defect reported in the mettl5 mutant. In this revision, the authors have added a more thorough analysis of clock gene expression and show that PER protein levels are increased relative to wild type animals a specific times of day, indicating increased stability of the protein. Given that PER inhibits its own transcription, the per RNA is low in the mutants. Efforts toward a more detailed understanding of how clock gene expression was altered in the mutants, as well as other clarification of sleep phenotypes throughout is appreciated. As noted above, a strength of this work is its relevance to a human developmental disorder as well as the transcriptomic and ribosomal profiling of the mutant. However, there still remain some minor weaknesses in the manuscript. This reviewer is not in agreement with the interpretation of the epigenetic experiments. Specifically, co-expression of Clk[jrk] or per [01] with the mettl5 mutant recovered the nighttime sleep phenotype, but was additive to the daytime sleep phenotype such that double mutants showed higher sleep. This effect should be acknowledged and discussed. Overall, this is an interesting paper that indicates a molecular link between mettl5 and the circadian clock in regulation of sleep.
**Recommendations for the authors:**

**Reviewer #3 (Recommendations for the authors):**
The authors misunderstood my original comment for Fig 1A. Please provide an explanation for the significance of the boxed region. There is little or no detail in the legend to help guide the reader.

The information has been added to the figure legends for Figure 1A.

Efforts toward improving analysis of circadian genes as well as sleep phenotypes (sleep onset time, rebound, etc) is much appreciated, thank you. However, Figure S1H and G panel labels are mixed up; please label in the order that they appear and that they correspond to the main text. Why is Figure S1H labeled "ZT 14"?

Sleep latency is defined as the time from preparing to sleep to actually falling asleep. In this study, it specifically refers to the time taken for each individual fly to reach the sleep phenotype (i.e., 25 minutes of continuous sleep). We noted that this label was misleading, as the actual time to reach the sleep phenotype varied among individual flies. Therefore, in the revised figures, we have removed the ZT14 label. In addition, we have corrected the labeling of Figures S1G and S1H to ensure they appear in the correct order and correspond accurately to the descriptions in the main text.

Unfortunately, based on Fig S1A-C, I am not convinced that mettl5 localizes to neurons, as there are no cells that show double labelling. This figure does not support the statement: "we found expression in both neurons (colocalizing with ELAV staining: Figure S1A-C) (lines 91-92), and "Mettl5-Gal4 is expressed in distinct neurons and glia that appear crucial for sleep regulation." (line 297). What "distinct" sleep related neurons were labeled? The staining in Fig S1A shows a different distribution from that in Fig S1D, and so it's possible this was a technical issue. Is there a better example?

Thank you for your careful review and valuable comments. We agree that the colocalization of METTL5 with the neuronal marker ELAV is relatively sparse. However, as indicated by the arrows in Fig S1A–C, we did observe a few cells showing clear double labeling. These examples support the presence of METTL5 expression in neurons, albeit at a low frequency.

In Figure 4G-H, please indicate the time of day of tissue collection.

In Figure 4G-H, the tissue was collected at ZT0. We have now indicated this time point in the figure and legend to clarify the experimental timing.

As noted in the public comment, I remain in disagreement with the assessment that "the double mutant showed the similar phenotype as downstream genes". The striking significant increase in daytime sleep in the double mutants remains unexplained. No further experiments are necessary, but this should be acknowledged in the text. Instead of an epistatic effect, given that overall sleep is high in the double mutants, another possible explanation is that the flies are sick and so are less active and sleeping more.

Thank you for your suggestion. This has been acknowledged in the text. “Genetic epistasis experiments further supported this model, with clock gene mutants modified *Mettl5* mutant phenotypes that suggesting both Clock and Per downstream of *Mettl5* (Figure 4I-N, Table 1). Secondary effect may exist for the significant increase in daytime sleep in the double mutants.”